# Lip movements entrain the observers' low-frequency brain oscillations to facilitate speech intelligibility

**Hyojin Park\*, Christoph Kayser, Gregor Thut, Joachim Gross\***

Institute of Neuroscience and Psychology, University of Glasgow, Glasgow, United Kingdom

**Abstract** During continuous speech, lip movements provide visual temporal signals that facilitate speech processing. Here, using MEG we directly investigated how these visual signals interact with rhythmic brain activity in participants listening to and seeing the speaker. First, we investigated coherence between oscillatory brain activity and speaker's lip movements and demonstrated significant entrainment in visual cortex. We then used partial coherence to remove contributions of the coherent auditory speech signal from the lip-brain coherence. Comparing this synchronization between different attention conditions revealed that attending visual speech enhances the coherence between activity in visual cortex and the speaker's lips. Further, we identified a significant partial coherence between left motor cortex and lip movements and this partial coherence directly predicted comprehension accuracy. Our results emphasize the importance of visually entrained and attention-modulated rhythmic brain activity for the enhancement of audiovisual speech processing.

**\*For correspondence:** Hyojin. Park@glasgow.ac.uk (HP); Joachim.Gross@glasgow.ac.uk (JG)

**Competing interests:** The authors declare that no competing interests exist.

## Introduction

Communication is one of the most fundamental and complex cognitive acts humans engage in. In a dialogue, a large range of dynamic signals are exchanged between interlocutors including body posture, (emotional) facial expressions, head and eye movements, gestures and a rich acoustic speech signal. Movements of the lips contain sufficient information to allow trained observers to comprehend speech through visual signals. Even for untrained observers and in the presence of auditory signals, lip movements can be beneficial for speech comprehension when the acoustic signal is degraded (*Peelle and Sommers, 2015*; *Sumby and Pollack, 1954*; *van Wassenhove et al., 2005*; *Zion-Golumbic and Schroeder, 2012*). Dynamic lip movements support disambiguation of syllables and can provide temporal onset cues for upcoming words or syllables (*Chandrasekaran et al., 2009*; *Grant and Seitz, 2000*; *Kim and Davis, 2003*; *Schroeder et al., 2008*; *Schwartz and Savariaux, 2014*).

However, it has remained unclear how dynamic lip movements during continuous speech are represented in the brain and how these visual representations interact with the encoding of the acoustic speech signal. A potential underlying mechanism for the visual enhancement of hearing could be the synchronization of brain rhythms between interlocutors, which has been implicated in the encoding of acoustic speech (*Giraud and Poeppel, 2012*; *Hasson et al., 2012*; *Pickering and Garrod, 2013*). Indeed, continuous speech and the associated lip movements show temporal modulations at the syllabic rate (3–8 Hz) (*Chandrasekaran et al., 2009*). These signals produced in the speaker's motor system supposedly lead to resonance in the listener's brain that facilitates speech comprehension (*Giraud and Poeppel, 2012*). The hallmark of such a process is the synchronization of brain activity at the frequency of dominant rhythmic components in the communication signal (*Schroeder et al.,*

**eLife digest** People are able communicate effectively with each other even in very noisy places where it is difficult to actually hear what others are saying. In a face-to-face conversation, people detect and respond to many physical cues – including body posture, facial expressions, head and eye movements and gestures – alongside the sound cues. Lip movements are particularly important and contain enough information to allow trained observers to understand speech even if they cannot hear the speech itself.

It is known that brain waves in listeners are synchronized with the rhythms in a speech, especially the syllables. This is thought to establish a channel for communication – similar to tuning a radio to a certain frequency to listen to a certain radio station. Park et al. studied if listeners' brain waves also align to the speaker's lip movements during continuous speech and if this is important for understanding the speech.

The experiments reveal that a part of the brain that processes visual information – called the visual cortex – produces brain waves that are synchronized to the rhythm of syllables in continuous speech. This synchronization was more precise in a complex situation where lip movements would be more important to understand speech. Park et al. also found that the area of the observer's brain that controls the lips (the motor cortex) also produced brain waves that were synchronized to lip movements. Volunteers whose motor cortex was more synchronized to the lip movements understood speech better. This supports the idea that brain areas that are used for producing speech are also important for understanding speech.

Future challenges include understanding how synchronization of brain waves with the rhythms of speech helps us to understand speech, and how the brain waves produced by the visual and motor areas interact.

*2008*). Consistent with this idea, previous studies have demonstrated the frequency-specific synchronization between brain activity and continuous auditory speech signals (*Ahissar et al., 2001*; *Ding and Simon, 2012*; *Gross et al., 2013b*; *Luo and Poeppel, 2007*; *Peelle et al., 2013*) at frequencies below 10 Hz. This synchronization was found to be stronger for intelligible than non-intelligible speech and facilitated by top-down signals from left inferior frontal and motor areas (*Ding and Simon, 2014*; *Kayser et al., 2015*; *Park et al., 2015*) as well as during attention (*Zion-Golumbic and Schroeder, 2012*).

The correlated temporal dynamics of the acoustic and lip signals raise the possibility that lip-mediated benefits for hearing rely on similar entrainment mechanisms in the observer as the acoustic component. This is plausible as the auditory speech entrainment and speech intelligibility are enhanced when congruent visual speech is present (*Crosse et al., 2015*; *Zion Golumbic et al., 2013*). Still, the neural representations of dynamic lip signals and their dependence on attention and the acoustic speech component remain unclear.

Here we directly tested four hypotheses: First, we hypothesized that rhythmic components in visual speech entrain brain activity in the observer. Second, to test whether benefits arising from seeing the speaker's lip movements are mediated through mechanism other than those implicated in auditory entrainment, we asked whether and which brain areas synchronize to lip movements independently of auditory signals. Third, we hypothesized that the synchronization between visual speech and brain activity is modulated by attention and congruence of visual and auditory signals. Finally, we tested whether any observed synchronization is relevant for speech comprehension.

We recorded MEG signals while participants perceived continuous audiovisual speech. To dissociate the synchronization to attended and unattended visual and acoustic signals, we manipulated the congruency of visual and acoustic stimuli in four experimental conditions (*Figure 1A*).

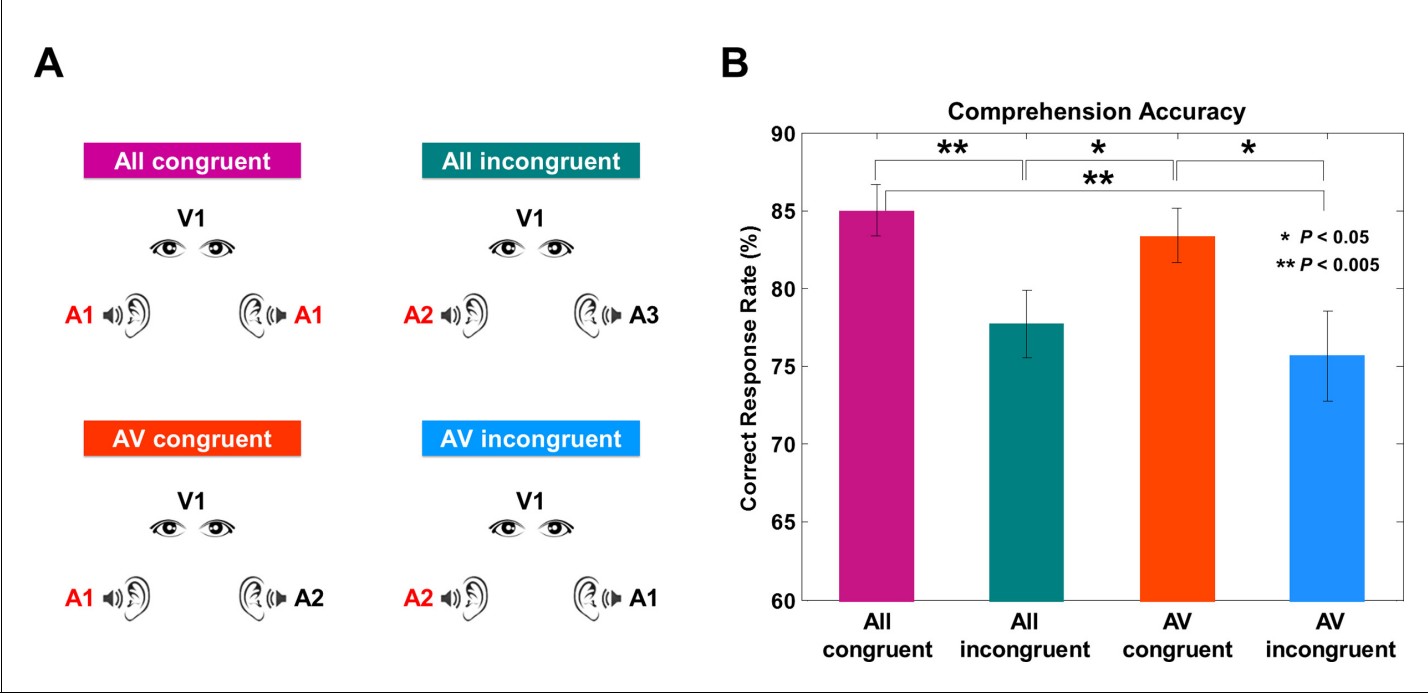

**Figure 1.** Experimental conditions and behavioral results. (**A**) Four experimental conditions. 'A' denotes auditory stimulus and 'V' denotes visual stimulus.The number refers to the identity of each talk. All congruent condition: Natural audiovisual speech condition where auditory stimuli to both ears and visual stimuli are congruent (from the same movie; A1, A1, V1). All incongruent condition: All three stimuli are from different movies (A2, A3, V4) and participants are instructed to attend to auditory information presented to one ear. AV congruent condition: Auditory stimulus presented to one ear matches the visual information (A5, A6, V5). Participants attend to the talk that matches visual information. AV incongruent condition: Auditory stimulus presented to one ear matches the visual information (A7, A8, V8). Participants attend to the talk that does not match the visual information. Attended stimulus is marked as red color for the group attended to the left side (see Materials and methods for details). (**B**) Behavioral accuracy by comprehension questionnaires. Congruent conditions show high accuracy rate compared to incongruent conditions (%; mean ± s.e.m.): All congruent: 85 ± 1.66, All incongruent: 77.73 ± 2.15, AV congruent: 83.40 ± 1.73, AV incongruent: 75.68 ± 2.88). Statistics between conditions show significant difference only between congruent and incongruent conditions (paired *t*-test, df: 43, p<0.05).

# Results

## Behavioral results

The four experimental conditions were designed to modulate congruence and informativeness of visual and auditory stimuli (*Figure 1A*). In each condition, one visual stimulus was presented and two (identical or different) speech streams were presented to the left and the right ears, respectively (see Materials and methods for details). The *All congruent* condition consisted of three congruent stimuli. The *All incongruent* condition had three incongruent stimuli. In the *AV congruent* condition, participants attended an auditory stimulus that had a congruent visual stimulus with an additional incongruent auditory stimulus. In the *AV incongruent* condition participants attended an auditory stimulus that was incongruent to a congruent audiovisual stimulus pair.

Overall, participants showed high comprehension accuracy across conditions (%; mean ± s.e.m.): All congruent: 85 ± 1.66, All incongruent: 77.73 ± 2.15, AV congruent: 83.40 ± 1.73, AV incongruent: 75.68 ± 2.88). As expected, accuracy was significantly higher when the visual stimulus was congruent with attended auditory stimulus (i.e., All congruent and AV congruent conditions) compared to when the visual stimulus was incongruent with attended auditory stimulus (i.e., All incongruent and AV incongruent conditions) (*Figure 1B*; paired *t*-test, df: 43, p<0.05; All congruent vs. All incongruent: *t* = 3.09, p=0.003, All congruent vs. AV congruent: *t* = 0.76, p=0.45 (n.s.), All congruent vs. AV incongruent: *t* = 2.98, p=0.004, AV congruent vs. All incongruent: *t* = 2.15, p=0.03, AV congruent vs. AV incongruent: *t* = 2.24, p=0.03, All incongruent vs. AV incongruent: *t* = 0.65, p=0.52 (n.s.)). Interestingly, performance for AV congruent condition was not significantly different to performance in All

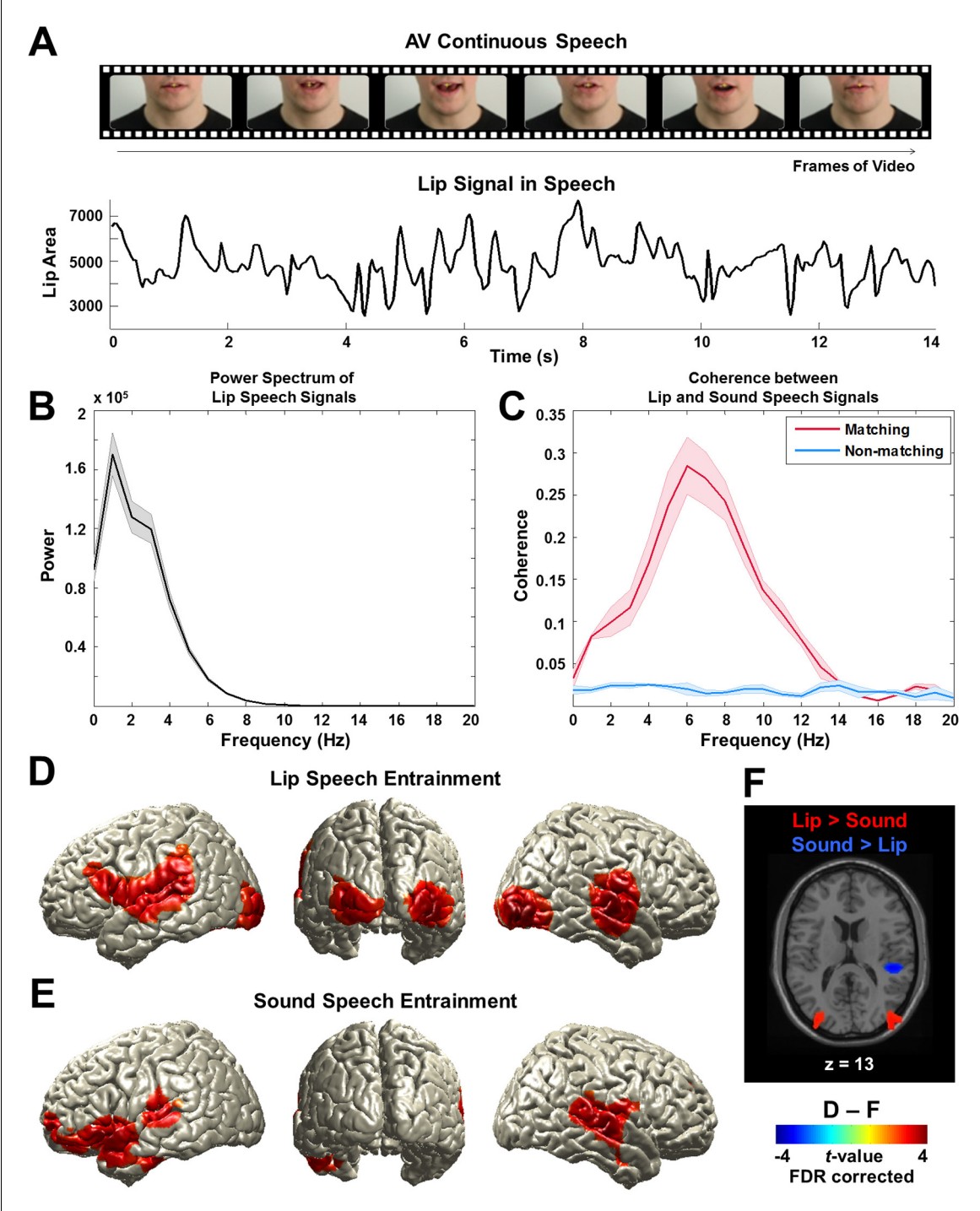

**Figure 2.** Lip signals in continuous speech and its entrainment in the brain. (A) Lip signals in the continuous audiovisual speech.Lip contour was extracted for each video frame and corresponding area was computed (see *Figure 2—figure supplement 1A,B,C* for details). One representative lip speech signal (for around 14 s speech) is shown here. Speaker's face is cropped for this publication only but not in the original stimuli. (B) Spectral profile of lip speech signals. The power spectra of lip speech signals used in this study were averaged (mean ± s.e.m.). Signal is dominated by low-frequency components from 0 to 7 Hz that robustly peak around 0 to 4 Hz corresponding to delta and theta band neuronal oscillations in the brain. (C) Coupling between lip and sound speech signals by coherence. Coherence between matching (red line) and non-matching (blue line) lip and sound speech signals were computed and averaged across talks used in the study (mean ± s.e.m.). (D) Lip speech entrainment in natural audiovisual speech (All congruent condition). Coherence was computed between lip speech signal and brain activity at each voxel and then statistically compared to surrogate data at 1 Hz (the dominant frequency in the power spectrum in (b); p<0.05, FDR corrected). (E) Sound speech entrainment in natural audiovisual speech (All congruent condition). Using sound speech envelope, the same computation described in (D) was performed to investigate

*Figure 2 continued on next page*

*Figure 2 continued*

sound speech entrainment effect (p<0.05, FDR corrected). (F) Lip speech- and sound speech-specific entrainment effects. Lip speech (D) and sound speech coherence (E) were statistically compared (p<0.05, FDR corrected).

The following figure supplements are available for figure 2:

**Figure supplement 1.** A schematic figure for the analysis of coupling between lip movements and brain activity.

**Figure supplement 2.** Brain activity entrained by lip movements.

**Figure supplement 3.** Brain activity entrained by lip movements and sound envelope.

congruent condition despite the interfering auditory input. This is likely caused by attentional efforts to overcome interfering input leading to behavioral compensation.

## Lip and sound signals are coherent during continuous speech

To examine the frequency spectrum of the lip signal, we computed the lip area for each video frame (*Figure 2A* and *Figure 2—figure supplement 1A,B,C*). The signal is dominated by low-frequency components from 0 to 7 Hz peaking around 0 to 4 Hz (*Figure 2B*; from all lip speech signals used in this study; mean ± s.e.m.). Next, we computed coherence between these lip signals and the respective acoustic signals to investigate the relationship between visual and auditory components in audiovisual speech. This was computed for all talks used in the study and averaged. The coherence spectrum reveals a prominent peak in a frequency band corresponding to the syllable rate around 4–8 Hz (red line; mean ± s.e.m.) (*Figure 2C*). These results demonstrate the temporal coupling of auditory and visual speech components.

## Lip movements during continuous speech entrain brain activity

First, we tested the hypothesis that lip movements entrain the observer's brain activity. We addressed this by computing coherence between the lip signal and brain signal at each voxel at frequencies ranging from 1 to 7 Hz (in 1 Hz steps) covering the spectral profile of the lip signals (*Figure 2B*). In addition, as a control, we computed surrogate maps (from time-shifted lip signals, thereby destroying physiologically meaningful coherence) as an estimate of spatially and spectrally specific biases of the analysis.

We first compared natural audiovisual speech condition (All congruent) and surrogate data for the frequency that showed strongest power in the lip signal (1 Hz). This revealed a significant entrainment effect in visual, auditory, and language areas bilaterally (p<0.05, false discovery rate (FDR) corrected; *Figure 2D*). The areas include early visual (V1; Calcarine sulcus) and auditory (A1; Heschl's gyrus) areas as well as inferior frontal gyrus (IFG; BA 44) (see *Figure 2—figure supplement 2* for the other conditions at 1 Hz).

However, since the speech envelope and lip movements are coherent (*Figure 2C*), it may be that this lip entrainment is induced by speech entrainment and not by lip movements per se. Thus, we performed the same coherence analysis for the sound speech envelope. In accordance with previous work (*Gross et al., 2013b*), we observed an extensive auditory network including Heschl's gyrus and superior/middle temporal gyri bilaterally and left frontal areas (p<0.05, FDR corrected; *Figure 2E*) (see *Figure 2—figure supplement 3* for different frequencies [2–5 Hz]). Statistical comparison of lip movement entrainment (*Figure 2D*) to sound speech entrainment (*Figure 2E*) revealed significantly stronger lip entrainment in bilateral visual areas and stronger sound speech coherence in right superior temporal gyrus (p<0.05, FDR corrected; *Figure 2F*). This demonstrates significant entrainment of brain activity to the lip movements irrespective of entrainment to the acoustic speech signal. In addition, we found significant lip movement entrainment in visual areas in the absence of a congruent auditory stimulus (*Figure 2—figure supplement 2B*). These results demonstrate for the first time the entrainment of cortical brain oscillations to lip movements during continuous speech.

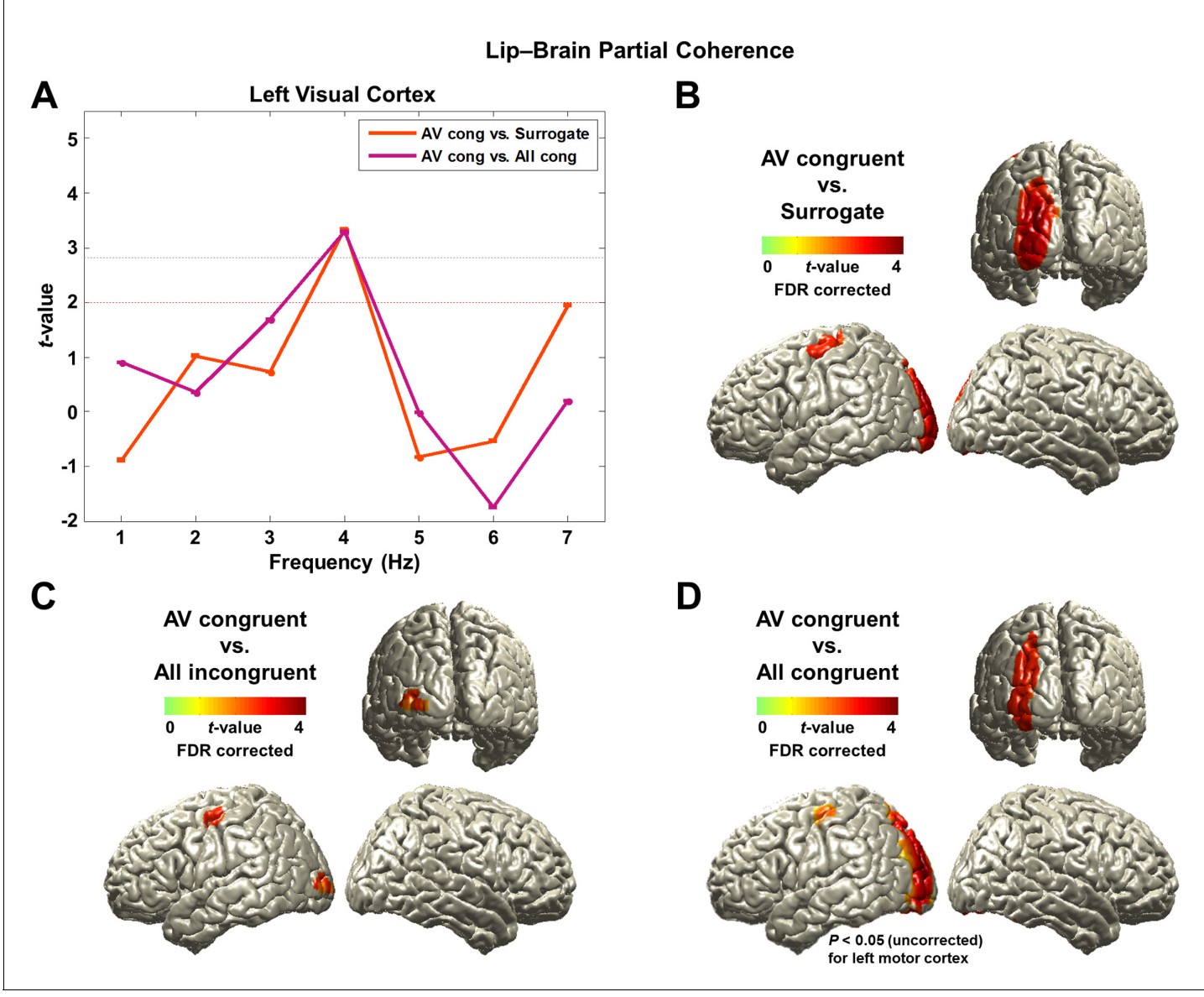

**Figure 3.** Lip-brain partial coherence. (A) Modulation of partial lip-brain coherence by attention and congruence in visual ROI.AV congruent condition was compared to the other conditions (paired *t*-test, df: 43, red dashed line: p<0.05, gray dashed line: p<0.05, corrected). (B, C, D) Attention-modulated partial coherence at each brain voxel (AV congruent versus surrogate (B), All incongruent (C), All congruent (D)). It shows significant involvement of left motor cortex (precentral gyrus; BA 4/6) and left visual areas (p<0.05, FDR corrected; but in (D), left motor cortex is observed at uncorrected p<0.05). Entrainment in the left motor cortex shows a systematic modulation such that statistical contrast with a strong difference in visual attention show stronger entrainment (AV congruent versus surrogate (B; $t_{43}$-value: 3.42) > All incongruent (C; $t_{43}$-value: 3.20) > All congruent (D; $t_{43}$-value: 2.24)).

The following figure supplements are available for figure 3:

**Figure supplement 1.** Lip-brain coherence.

**Figure supplement 2.** Partial coherence between lip movements and left motor cortex.

## Lip entrainment is modulated by attention and congruence

Next, we compared visual lip entrainment across conditions to test our hypothesis that entrainment changes with attention and the congruence of audiovisual stimuli. We focused our analysis on AV congruent condition where a distracting auditory speech stream is presented to one ear. Compared

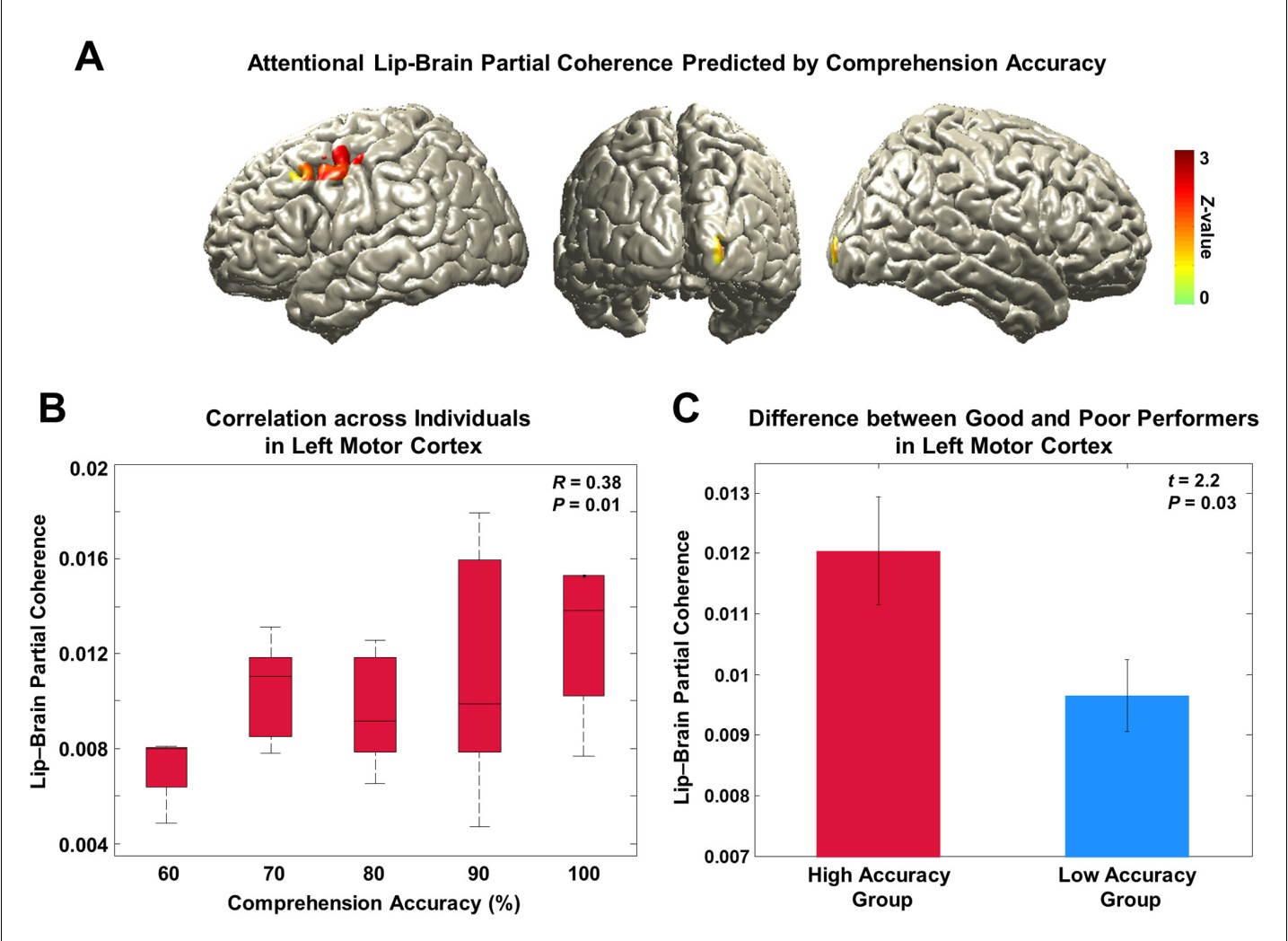

**Figure 4.** Behavioral correlates of attentional lip entrainment. (**A**) Lip-entrained brain regions predicted by attention-modulated comprehension accuracy. Regression analysis using comprehension accuracy across participants on the partial coherence map was performed at 4 Hz in each condition. Then Z-difference map was obtained from the regression analysis of conditions showing strongest difference in comprehension accuracy (AV congruent versus AV incongruent; see behavioral results in *Figure 1B*). This revealed that the left motor cortex entrainment predicts attention-modulated comprehension accuracy (Z-difference map at p<0.005). (**B**) Correlation between partial coherence in left motor cortex and comprehension accuracy. Partial coherence values from the maximum coordinate in the left motor cortex and comprehension accuracy across subjects for the AV congruent condition was positively correlated (Pearson's coefficient of Fisher's Z-transformed data $R = 0.38$, $P = 0.01$; Spearman rank correlation $R = 0.32$, $P = 0.03$). (**C**) Difference of attentional lip-entrainment in left motor cortex between good and poor performing group. Group t-statistics between good and poor performing group on the extracted partial coherence values in the left motor cortex for the AV congruent condition was performed. The two groups were divided using median value (90%; 23 good versus 21 poor performers) of comprehension accuracy for the AV congruent condition. Good performers showed higher partial coherence value in left motor cortex than poor performers (two-sample t-test on Fisher's Z-transformed data; $t_{42} = 2.2$, $P = 0.03$).

to the All congruent condition, AV congruent demands additional attention to visual speech because the visual signal is informative to disambiguate the two incongruent auditory streams. We therefore contrasted AV congruent with All congruent condition to capture the effect of visual attention. We also contrasted AV congruent condition with All incongruent condition to capture the effect of congruence.

Since lip and sound speech signals are coherent (*Figure 2C*), it is difficult to disentangle visual and auditory contributions to the lip movement entrainment. To measure lip-specific entrainment effects more directly, we computed partial coherence between lip movement signals and brain

activity while removing the contribution of acoustic speech signals. This provides an estimate of entrainment by lip movement signals that cannot be explained by acoustic speech signals and allowed us to test our second hypothesis that lip entrainment is not mediated via acoustic entrainment. However, we repeated the same analysis using coherence instead of partial coherence (*Figure 3—figure supplement 1*).

First, we identified the frequency band showing the strongest attention effect by averaging voxels across visual cortex (superior/middle/inferior occipital gyri) defined from AAL (Automated Anatomical Labeling) ROI (Region-of-Interest) map. We averaged the partial coherence values within the ROI and statistically compared the AV congruent to surrogate data and to All congruent condition at each frequency from 1 to 7 Hz (*Figure 3A*; paired *t*-test, df: 43, red dashed line: p<0.05, gray dashed line: p<0.05, corrected). This revealed significantly stronger lip movement entrainment at 4 Hz in the left visual cortex for AV congruent compared to both, All congruent condition and surrogate data. In AV congruent condition, lip movements are informative and assist comprehension. This result suggests that coupling of low-frequency brain activity to lip movements is enhanced by visual attention.

Next, we studied attentional lip movement entrainment at 4 Hz across the entire brain. This revealed a significant partial coherence between lip movements and left motor cortex (precentral gyrus; BA 4/6) in addition to the left visual areas (p<0.05, FDR corrected; *Figure 3B,C,D*). Entrainment in the left motor cortex shows a systematic modulation by attention. Specifically, contrasts with a stronger difference in visual informativeness exhibit stronger lip entrainment (AV congruent versus surrogate (*Figure 3B*; $t_{43}$-value: 3.42) > All incongruent (*Figure 3C*; $t_{43}$-value: 3.20) > All congruent (*Figure 3D*; $t_{43}$-value: 2.24)). Contrasting AV congruent with All congruent conditions (*Figure 3D*), which both have congruent visual speech, also revealed an effect in the same motor cortex area (p<0.05, uncorrected; for the frequency-specific plot for left motor cortex, see *Figure 3—figure supplement 2*). In the addition, the left visual cortex shows significantly stronger lip entrainment for AV congruent compared to surrogate data (*Figure 3B*), All incongruent condition (*Figure 3C*) and All congruent condition (*Figure 3D*). This demonstrates that activity in left motor cortex and left visual areas show significant alignment to lip movements independent of sound speech signals and that this alignment is stronger when visual speech is more informative and congruent to the auditory stimulus.

## Attentional enhancement of lip entrainment in left motor cortex facilitates speech comprehension

To address our fourth hypothesis that lip-entrained brain activity has an impact on speech comprehension, we identified brain regions where lip entrainment correlates with behavioral performance. We performed regression analysis using comprehension accuracy across subjects on the partial coherence map at 4 Hz in each condition. We then contrasted the condition with high visual attention and behavioral performance (AV congruent) to the condition with low visual attention and behavioral performance (AV incongruent; see behavioral results in *Figure 1B*). The regression *t*-values in the two conditions were transformed to standardized *Z*-value at each voxel and the two *Z*-maps were subtracted. This revealed that entrainment in left motor cortex predicts attention-modulated comprehension accuracy (*Figure 4A*; *Z*-difference map at p<0.005).

To confirm this effect, we correlated partial coherence values from the maximum coordinate in the left motor cortex with comprehension accuracy across subjects for the AV congruent condition. This revealed significant positive correlation demonstrating that individuals with higher levels of comprehension show higher partial coherence in the left motor cortex. Since the accuracy has discrete values such as 70, 80, 90% (correct response rate out of 10 questions at the comprehension testing), we plotted the correlation as a box plot (*Figure 4B*; Pearson's coefficient of Fisher's *Z*-transformed data *R* = 0.38, *P* = 0.01; Spearman rank correlation *R* = 0.32, *P* = 0.03).

To confirm this result in a separate analysis, we performed *t*-statistics between good and poor performing group on the extracted partial coherence values in the AV congruent condition. The two groups were divided using median value of performance leading to 23 good performers and 21 poor performers. Good performers showed significantly higher partial coherence value in the left motor cortex than poor performers (*Figure 4C*; two-sample *t*-test on Fisher's *Z*-transformed data; $t_{42}$ = 2.2, *P* = 0.03). Taken together, these results demonstrate that stronger attentional lip entrainment in the left motor cortex supports better speech comprehension.

## Discussion

Here we provide the first direct evidence that lip movements during continuous speech entrain low-frequency brain oscillations in speech processing brain areas, that this entrainment is modulated by attention and congruence, and that entrainment in motor regions correlates with speech comprehension.

### Speaking lips entrain low-frequency oscillations during natural audiovisual speech

Recent studies provide converging evidence that detailed information about the identity or specific features of sensory stimuli can be decoded from the low-frequency phase of LFP or MEG/EEG signals (*Ng et al., 2013*; *Panzeri et al., 2015*; *Schyns et al., 2011*). For example, a recent study has demonstrated that the identity of different audiovisual movie stimuli can be decoded from delta-theta phase in occipital MEG sensors (*Luo et al., 2010*). In the auditory domain, this stimulus-specificity is at least partly due to a phase synchronization of rhythmic brain activity and the auditory speech envelope (*Gross et al., 2013b*; *Peelle et al., 2013*). Recent studies have shown that congruent visual stimulation facilitates auditory speech entrainment (*Crosse et al., 2015*; *Zion Golumbic et al., 2013*). Here, we extend these results by showing low-frequency phase synchronization between speakers' lip movements and listeners' brain activity. This visual entrainment is clearly distinct from auditory entrainment for three reasons. First, some areas show stronger entrainment to the visual compared than to the auditory stimulus. Second, removing the auditory contribution using partial coherence still results in significant visual entrainment. Third, we report visual entrainment in the absence of a congruent auditory stimulus. Together, this establishes the existence of a visual entrainment mechanism for audiovisual speech in addition to the well-studied auditory entrainment. In correspondence to the auditory domain, this effect relies on the quasi-rhythmic nature of visual speech that is particularly prominent in the subtle but salient lip movements.

Our coherence analysis reveals an extensive network comprising speech processing areas such as bilateral primary sensory areas (V1 (Calcarine sulcus; BA 17) and A1 [Heschl's gyrus; BA 41]), extended sensory visual (BA 18/19) and auditory areas (superior/middle temporal gyri; BA 21/22/42), and posterior superior temporal sulcus (pSTS) as well as inferior frontal gyrus (IFG; BA 44/6). Although these areas show synchronization to lip movements, there is large overlap with areas showing synchronization to sound envelope. This is not surprising because both stimulus signals are coherent. But as expected, in occipital areas visual entrainment is significantly stronger compared to auditory entrainment.

An interesting asymmetry emerges in auditory areas in the comparison of lip entrainment to speech entrainment (*Figure 2F*). The right superior temporal cortex is significantly stronger coupled to speech envelope than to lip movements while there is no significant difference in left superior temporal cortex. This seems to suggest that visual speech predominantly entrains left temporal areas (in addition to visual areas). This is consistent with a recent fMRI study demonstrating preferential processing of visual speech in left superior temporal areas (*Blank and von Kriegstein, 2013*).

In summary, we find that visual areas are entrained by lip movements and left and right auditory areas by lip movements and speech envelope with only the right auditory cortex showing a stronger coupling to speech envelope compared to lip movements.

### Attention and congruence modulate audiovisual entrainment

We studied the effect of attention in an ecologically valid scenario with congruent audiovisual speech when a distracting auditory stimulus was present (AV congruent condition) or absent (AII congruent condition). While we did not explicitly manipulate attention in our paradigm, the distracting auditory stimulus in the AV congruent condition renders the visual speech relevant and informative. Attention to visual speech is known to help to disambiguate the competing auditory inputs (*Sumby and Pollack, 1954*; *Zion Golumbic et al., 2013*). This condition was compared to the AII congruent condition where attention to the visual stimulus is not required for speech comprehension.

Interestingly, attention to lip movements leads to significantly increased coupling in left hemispheric visual areas. Previous studies have demonstrated that observers process audiovisual speech preferentially based on information from the left side of a speaker's face (observers' right visual

hemifield) as compared to the right side (*Behne et al., 2008*; *Campbell et al., 1996*; *Smeele et al., 1998*; *Swerts and Krahmer, 2008*) (but see *Nicholls and Searle, 2006*). This has been related to an attentional bias to the observers' right visual hemifield due to the left hemisphere dominance for speech processing (*Thompson et al., 2004*). Since the right visual hemifield is represented in the left visual cortex, this attentional bias could explain the observed lateralization of entrainment.

While visual areas show both coherence and partial coherence to lip movements, temporal areas only emerge from coherence analysis (*Figure 2D* and *Figure 3—figure supplement 1*). This seems to suggest that the temporal alignment of auditory cortex activity to lip movements is not independant of the congruent acoustic speech stimulus and therefore speaks against a direct entrainment of auditory cortex activity by lip movements. However, the informativeness of lip movements still modulates significantly the coherence between lip movements and right temporal brain areas (*Figure 3—figure supplement 1B,C,D*) and indicates an indirect effect of visual attention. Similarly, left posterior superior temporal sulcus (pSTS) shows a significant effect of congruence (*Figure 3—figure supplement 1D*) that is only evident in coherence and not in partial coherence map. This is consistent with previous reports of pSTS as a locus of multisensory integration (*Hocking and Price, 2008*; *Noesselt et al., 2012*) with particular relevance for visual speech recognition (*Beauchamp et al., 2004*; *Blank and von Kriegstein, 2013*; *Werner and Noppeney, 2010*).

## Motor areas are entrained by speaking lips

Attended audiovisual speech leads to significant entrainment in left inferior precentral gyrus corresponding to the lip representation in motor cortex (*Figure 3—figure supplement 1C,D*) (*Giraud et al., 2007*). In addition, partial coherence analysis where the effect of speech envelope on lip entrainment is removed reveals a more superior left lateralized motor area (*Figure 3B,C,D*). Partial coherence in this area is modulated by congruence (*Figure 3C*) and attention (*Figure 3D*) and predicts comprehension accuracy (*Figure 4A*). There is ample evidence for the activation of left motor cortex during audiovisual speech perception (*Evans and Davis, 2015*; *Meister et al., 2007*; *Mottonen et al., 2013*; *Watkins et al., 2003*; *Wilson et al., 2004*; *Ylinen et al., 2015*). Our results indicate that left motor areas are not only activated but also entrained by audiovisual speech. This establishes a precise temporal coupling between audiovisual sensory inputs and sensory and motor areas that could temporally coordinate neuronal computations associated with speech processing. As part of a proposed dorsal auditory pathway, motor areas could provide access to an internal forward model of speech that closely interacts with auditory sensory systems (*Bornkessel-Schlesewsky et al., 2015*; *Rauschecker and Scott, 2009*). During speech production, an efference copy is sent to auditory areas to allow for efficient monitoring and control. During speech perception, sensory signals could be used to constrain the forward model in simulating the speaker's motor program and predicting upcoming sensory events (*Arnal and Giraud, 2012*; *Bornkessel-Schlesewsky et al., 2015*; *Lakatos et al., 2013*).

Indeed, a recent study has demonstrated direct top-down control of left auditory cortex from left motor cortex during continuous speech processing. The strength of low-frequency top-down signals in the left motor cortex correlates with the coupling of auditory cortex and sound speech envelope (*Park et al., 2015*) (see also *Kayser et al., 2015*). This suggests that the motor cortex plays a predictive role in speech perception consistent with recent demonstrations of its contribution to the temporal precision of auditory speech perception (*Morillon et al., 2014*; *2015*; *Wilson et al., 2008*).

Visual speech *per se* is not critical for speech comprehension, however, it facilitates auditory speech processing as it aids temporal prediction and can prime the auditory system for upcoming concordant auditory input (*Peelle and Sommers, 2015*; *van Wassenhove et al., 2005*). Overall, left motor cortex seems to be an important area for facilitating audiovisual speech processing through predictive control in auditory active sensing (*Meister et al., 2007*; *Morillon et al., 2015*).

In summary, our study provides the first direct evidence that lip movements during continuous speech entrain visual and motor areas, and that this entrainment is modulated by attention and congruence and is relevant for speech comprehension. This adds to similar findings in the auditory domain and provides a more comprehensive view of how temporally correlated auditory and visual speech signals are processed in the listener's brain. Overall, our findings support an emerging model where rhythmic audiovisual signals entrain multisensory brain areas and dynamically interact with an internal forward model accessed via the auditory dorsal stream to form dynamically updated predictions that improve further sensory processing. Through these mechanisms brain oscillations might

implement inter-subject synchronization and support the surprising efficiency of inter-human communication.

## Materials and methods

### Participants

Data were obtained from 46 healthy subjects (26 females; age range: 18–30 years; mean age: 20.54 ± 2.58 years) and they were all right-handed confirmed by Edinburgh Handedness Inventory (*Oldfield, 1971*). All participants provided informed written consent before participating in the experiment and received monetary compensation for their participation. All participants had normal or corrected-to-normal vision and normal hearing. None of the participants had a history of developmental, psychological, or neurological disorders. Only native English-speaking volunteers with British nationality were recruited due to the British accent in the stimulus material. Two subjects were excluded from the analysis (one subject fell asleep and one had MEG signals with excessive noise). This left dataset from 44 participants (25 females; age range: 18–30 years; mean age: 20.45 ± 2.55 years). This study was approved by the local ethics committee (CSE01321; University of Glasgow, College of Science and Engineering) and conducted in conformity with the Declaration of Helsinki.

### Data acquisition

Neuromagnetic signals were obtained with a 248-magnetometers whole-head MEG (Magnetoencephalography) system (MAGNES 3600 WH, 4-D Neuroimaging) in a magnetically shielded room using a sampling rate of 1017 Hz. MEG signals were denoised with information from the reference sensors using the denoise_pca function in FieldTrip toolbox (*Oostenveld et al., 2011*). Bad sensors were excluded by visual inspection. Electrooculographic (EOG) and electrocardiographic (ECG) artifacts were eliminated using independent component analysis (ICA). Participants' eye fixation and movements were recorded during the experiment using an eye tracker (EyeLink 1000, SR Research Ltd.) to ensure that they fixate on the speaker's lip.

T1-weighted structural magnetic resonance images (MRI) were acquired at 3 T Siemens Trio Tim scanner (Siemens, Erlangen, Germany) with the following parameters: 1.0 x 1.0 x 1.0 mm$^3$ voxels; 192 sagittal slices; Field of view (FOV): 256 x 256 matrix. Data will be available upon request by contacting corresponding authors.

### Stimuli and Experiment

The stimuli used in this study were audiovisual video clips of a professional male speaker talking continuously (7–9 min). The talks were originally taken from TED talks (www.ted.com/talks/) and edited to be appropriate to the stimuli we used (e.g. editing words referring to visual materials, the gender of the speaker).

Eleven video clips were filmed by a professional filming company with high-quality audiovisual device and recorded in 1920 x 1080 pixels at 25 fps (frame per second) for video and sampling rate of 48 kHz for audio.

In a behavioral study, these videos were rated by 33 participants (19 females; aged 18–31 years; mean age: 22.27 ± 2.64 years) in terms of *arousal, familiarity, valence, complexity, significance (informativeness), agreement (persuasiveness), concreteness, self-relatedness, and level of understanding.* Participants were instructed to rate each talk on these 9 items using Likert scale (*Likert, 1932*) 1 to 5 (for an example of concreteness, 1: very abstract, 2: abstract, 3: neither abstract nor concrete, 4: concrete, 5: very concrete). Talks with excessive mean scores (below 1 and over 4) were excluded and 8 talks were selected for the experiment.

There were four experimental conditions: All congruent, All incongruent, AV congruent, AV incongruent (*Figure 1A*). In each condition, one video recording was presented and two (identical or different) auditory recordings were presented to the left and the right ear, respectively.

All congruent condition. Natural audiovisual speech condition where auditory stimuli to both ears and visual stimuli are congruent (from the same movie; e.g. A1, A1, V1 – where the first A denotes stimulus presented to the left ear, second A denotes stimulus presented to right ear and V denotes visual stimulus. The number refers to the identity of each talk.).

All incongruent condition. All three stimuli are from different movies (e.g. A2, A3, V4) and participants are instructed to attend to auditory information presented to one ear.

AV congruent condition. Auditory stimulus presented to one ear matches the visual information (e.g. A5, A6, V5). Participants attend to the talk that matches visual information.

AV incongruent condition. Auditory stimulus presented to one ear matches the visual information (e.g. A7, A8, V8). Participants attend to the talk that does not match the visual information.

The color (yellow or blue) of a small fixation cross which was overlaid on the speaker's lip indicates the side of attention (left or right talk, e.g. "If the color of fixation cross is yellow, please attend to left ear talk."). The functional meaning of the color of fixation cross was counterbalanced across subjects. In All congruent condition (natural audiovisual speech), participants were instructed to ignore the color and just to attend to both sides. Participants were instructed to fixate on the speaker's lip all the time in all experimental conditions even if they found it difficult to do so in some conditions (e.g. incongruent conditions; All incongruent and AV incongruent). The fixation cross was used to guide participants' fixation on the speaker's lip. Furthermore, the gaze behavior was monitored by an eye tracker. The importance of eye fixation on the speaker's lip was stressed at the task instruction and they were notified that their eye fixation would be monitored by an eye tracker.

There were two groups (22 subjects each). Participants in one group attended to left ear talk and participants in another group attended to right ear talk in the experiment (for all four conditions). Since the attended side is the same within subjects, in order to avoid presenting all the same color of fixation cross (e.g. yellow) within subjects and to prevent them from sensing that they always attend to one side (left or right), the All congruent condition was always presented in the middle (second or third) among the four conditions using color of fixation cross indicating opposite side (e.g. blue). As expected there was no significant difference in comprehension accuracy between groups (two sample $t$-test, df: 42, p>0.05) and for the questions addressed here data was pooled across both groups.

In order to prevent talk-specific effects, we used two sets of stimuli consisting of different talks in the combination of audiovisual talks and these two sets were randomized across subjects (each set 1 and 2 was used for a half of subjects (22 subjects)). For example, talks used for AV incongruent condition in the set 1 were used for All incongruent condition in the set 2.

To assess the level of comprehension, we designed questionnaires for each talk. Each questionnaire consists of 10 questions about the talk and tests general comprehension of the talk (e.g., "What is the speaker's job?"). These questionnaires were also validated from another set of participants (16 subjects; 13 females; aged 18–23 years; mean age: 19.88 ± 1.71 years) to ensure the same level of difficulty (accuracy) across questionnaires for the talks and the length (word count) of the questionnaires also matched across all the questionnaires. The attended 4 talks (in the 4 conditions) were counterbalanced across conditions in the two sets, thus comprehension of 4 talks were validated. Three participants who showed poor performance (below 60% accuracy) were excluded from the analysis. There were no significant differences in the comprehension accuracy between the talks (mean ± s.e.m. accuracy (%) for talk 1: 84.62 ± 2.91; talk 2: 87.69 ± 2.31; talk 3: 90.77 ± 2.39; talk 4: 85.38 ± 2.43; p>0.05 at all pair-wise $t$-tests between talks).

In order to recombine audiovisual talks for the four experimental conditions and to add fixation cross, we used Final Cut Pro X (Apple Inc., Cupertino, CA).

The stimuli were controlled with Psychtoolbox (*Brainard, 1997*) under MATLAB (MathWorks, Natick, MA). Visual stimuli were delivered with a resolution of 1280 x 720 pixels at 25 fps (mp4 format). Auditory stimuli were delivered at 48 kHz sampling rate via a sound pressure transducer through two 5 meter-long plastic tubes terminating in plastic insert earpieces.

Each condition (continuous audiovisual speech) lasted 7–9 min. After each condition, comprehension questionnaire was performed about the attended talk. Here we measured both accuracy and response time and participants were asked to respond as accurately and quickly as possible. After the experiment, post-experimental questionnaire was administered to obtain participants' feedback about the experiment.

## Data analysis

The analysis of MEG data was performed using the FieldTrip toolbox (*Oostenveld et al., 2011*) and in-house MATLAB codes according to guidelines (*Gross et al., 2013a*).

## Lip speech (visual) signal processing

We used in-house Matlab code to extract lip contour of the speaker for each frame of each movie (*Figure 2A*). From the lip contour we computed area information (area within lip contour), major axis information (horizontal axis within lip contour) and minor axis information (vertical axis within lip contour). For our analysis we used area information of lip contour (see *Figure 2—figure supplement 1A,B,C* for details) although use of vertical axis leads to qualitatively similar results. This signal was resampled at 250 Hz to match the sampling rate of the preprocessed MEG signal.

## Sound speech (auditory) signal processing

We computed the amplitude envelope of sound speech signals (*Chandrasekaran et al., 2009*). We constructed eight frequency bands in the range 100–10,000 Hz to be equidistant on the cochlear map (*Smith et al., 2002*). Then sound speech signals were band-pass filtered in these bands using a fourth-order Butterworth filter (forward and reverse). Hilbert transform was applied to obtain amplitude envelopes for each band. These signals were then averaged across bands and resulted in a wideband amplitude envelope. Finally, these signals were downsampled to 250 Hz for further analysis.

## MEG-MRI co-registration

MR image of each participant was co-registered to the MEG coordinate system using a semi-automatic procedure. Anatomical landmarks such as nasion, bilateral pre-auricular points were manually identified in the individual's MRI. Based on these three points, both coordinate systems were initially aligned. Subsequently, numerical optimization was achieved by using the ICP algorithm (*Besl and McKay, 1992*).

## Source localization

Individual head model was created from structural MRI using segmentation routines in FieldTrip and SPM8. Leadfield computation was based on a single shell volume conductor model (*Nolte, 2003*) using a 8-mm grid defined on the MNI (Montreal Neurological Institute) template brain. For spatial normalization to the standard template, the template grid was transformed into individual head space by linear spatial transformation. Cross-spectral density matrices were computed using Fast Fourier Transform on 1-s segments of data after applying multitaper. Source localization was performed using DICS (*Gross et al., 2001*) and beamformer coefficients were computed sequentially for the frequency range from 1–10 Hz.

## Coherence between lip movement signals and MEG signals

In this study, we used coherence as a frequency-domain measure of dependency to study how rhythmic components in lip movements in continuous speech entrain neuronal oscillations.

First, frequency-specific brain activation time-series were computed by applying the beamformer coefficients to the MEG data filtered in the same frequency band (fourth order Butterworth filter, forward and reverse, center frequency ± 3 Hz). The lip speech signals were filtered in the same frequency band. Then, coherence was computed (*Rosenberg et al., 1989*) between the lip speech signal and source-localized brain signal for each voxel and each frequency band across 1-s-long data segments overlapping by 0.5 s (*Figure 2—figure supplement 1D*). This computation resulted in a volumetric map describing lip-entrained brain oscillations for each frequency band in each individual. This computation was performed for all experimental conditions: All congruent, All incongruent, AV congruent, AV incongruent.

In addition, surrogate maps were created by computing coherence between brain signals and 30 s-shifted lip speech signals for each of the four experimental conditions, thus destroy existing temporal dependencies between the two signals. This surrogate data serves as control data as computations use the same data after controlled manipulation that destroys the effect of interest (here the temporal shift removes temporal dependencies in coherence measure). Surrogate data therefore provide an estimate of coherence that can be expected by chance for each condition.

## Partial coherence

In audiovisual speech, auditory and visual inputs are coherent (*Chandrasekaran et al., 2009*) (also shown in *Figure 2C*). The main purpose of this study was to investigate whether visual signals (lip movements) entrain/modulate brain activity and where this entrainment occurs. In order to rule out functional coupling (coherence) explained by auditory signals, here we additionally computed partial coherence (*Rosenberg et al., 1998*), i.e., the coherence partialling out sound speech signals. The analysis process was identical to the process for the coherence above. This partial coherence provides entrained brain activity explained by lip movements that cannot be accounted for by auditory speech signal.

## Coherence between lip movements and speech signals

As explained above, in natural audiovisual speech, auditory and visual information are robustly correlated (*Chandrasekaran et al., 2009*). Here we computed the coherence between lip speech (visual) and sound speech (auditory) signals in all experimental conditions except All incongruent (All incongruent does not have matching lip-sound signals). Further, we computed coherence between non-matching lip and sound signals in all conditions except All congruent (All congruent does not have non-matching lip-sound signals).

## Statistics

Statistical analysis was performed on the data of all 44 participants using non-parametric randomization statistics in FieldTrip (Monte Carlo randomization). Specifically, individual volumetric maps were smoothed with a 10-mm Gaussian kernel and subjected to dependent samples *t*-test. We compared each condition with corresponding surrogate data and other experimental conditions. The null distribution was estimated using 500 randomizations and multiple comparison correction was performed using FDR (False Discovery Rate) (*Genovese et al., 2002*). Only significant results ($p < 0.05$, FDR corrected) are reported.

## Regression analysis using behavioral data

To study the relationship between lip movement entrainment and behavioral performance, we performed regression analysis across subjects using comprehension accuracy from each individual as regressors on the partial coherence map. In this regression analysis, we detected brain regions positively predicted by comprehension accuracy. To maximize sensitivity of this analysis, we compared AV congruent condition to the condition that showed strongest difference in behavioral performance – AV incongruent (*Figure 1B*). We performed the regression analysis for each condition, then *t*-values at each brain voxel from the regression analysis were transformed to standard *Z*-values to be compared between conditions. Then the *Z*-values were subtracted between the two conditions ($p < 0.005$).

## Acknowledgements

JG is supported by the Wellcome Trust (098433). TG is supported by the Wellcome Trust (098434). CK is supported by the UK Biotechnology and Biological Sciences Research Council (BBSRC; grant No BB/L027534/1) and the European Research Council (ERC-2014-CoG; grant No 646657). The funders had no role in study design, data collection and analysis, decision to publish, or preparation of the manuscript.

## Additional information

### Funding

| Funder | Grant reference number | Author |
|---|---|---|
| Biotechnology and Biological Sciences Research Council | BB/L027534/1 | Christoph Kayser |
| European Research Council | 646657 | Christoph Kayser |

| Wellcome Trust | 098433 | Gregor Thut Joachim Gross |
|---|---|---|
| Wellcome Trust | 098434 | Gregor Thut Joachim Gross |

The funders had no role in study design, data collection and interpretation, or the decision to submit the work for publication.

## Author contributions

HP, Conception and design, Acquisition of data, Analysis and interpretation of data, Drafting or revising the article, Contributed unpublished essential data or reagents; CK, GT, Conception and design, Analysis and interpretation of data, Drafting or revising the article; JG, Conception and design, Analysis and interpretation of data, Drafting or revising the article, Contributed unpublished essential data or reagents

## Author ORCIDs

Hyojin Park, http://orcid.org/0000-0002-7527-8280
Christoph Kayser, http://orcid.org/0000-0001-7362-5704
Joachim Gross, http://orcid.org/0000-0002-3994-1006

## Ethics

Human subjects: This study was approved by the local ethics committee (CSE01321; University of Glasgow, College of Science and Engineering) and conducted in conformity with the Declaration of Helsinki. All participants provided informed written consent before participating in the experiment and received monetary compensation for their participation.

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
