## [Decision Letter]

Thank you for submitting your article "Lip movements during speech entrain observers' low-frequency brain oscillations" for consideration by *eLife*. Your article has been favorably evaluated by Jody Culham as the Senior editor and three reviewers, one of whom is a member of our Board of Reviewing Editors.

The reviewers have discussed the reviews with one another and the Reviewing Editor has drafted this decision to help you prepare a revised submission.

Summary of the work:

This study examines the role of visual input during speech comprehension. The authors use MEG to show that cortical activity is entrained to lip movements, and that the synchronized activity is modulated by the congruence between visual and auditory inputs and possibly by attention. In addition to showing that entrainment in auditory and speech areas depends on the congruence of the auditory and visual speech signals, they found that activity in visual cortex and left motor cortex is synchronized to lip movements independently of auditory speech signals, and that the motor cortex coherence predicts comprehension accuracy. Although neural phase tracking (measured via coherence to external signals) in the low frequency domain during speech perception is a well-studied phenomenon, and previous work has demonstrated neural entrainment to visual stimuli, this study represents an important step forward by extending research in this field to the more ecologically valid condition of lip reading.

Essential revision requirements:

While the reviewers found the results compelling and concluded that they provide valuable insights into the way the brain responds to audiovisual speech, they identified the following concerns.

1) The reviewers felt that the paper would benefit from a stronger theoretical underpinning. At present, it is structured as an investigation into brain oscillations, which makes it hard to keep track of what are the main questions and findings. Some reorganization of the paper would help here.

2) The claim made at several points in the manuscript that lip movements entrain activity in auditory cortical areas does not appear to be justified. Although increased coherence was found there in the AV congruent vs. All incongruent condition, indicative of an interaction between the two modalities, lip movements in the absence of auditory speech (Figure 2—figure supplement 2) or with the contribution of speech signals removed (Figure 4) do not entrain auditory cortical activity. Thus, the effects on auditory cortex are only present during speech. The use of a partial correlation analysis to demonstrate that neural entrainment to lip movements is not an artifact caused by neural entrainment to the sound is very important and should be used throughout, not just toward the end of the Results.

3) Figure 2 shows results at 1 Hz while Figure 3 shows the contrast at 4 Hz. It is stated that Figure 2 focused on 1 Hz since significant entrainment was seen "especially at 1 Hz". Does that mean stronger coherence or broader activation? It would be helpful to show the coherence spectrum of representative areas or to show the source distribution at several frequencies. Additionally, for the source-space plots, showing the differences between the actual coherence values could be more informative.

4) Visual attention was not explicitly manipulated in this study. It is assumed that visual attention is enhanced when a competing sound is presented. Although this is a reasonable interpretation, it is not appropriate to report the auditory masking effect quite so definitively as a visual attention effect. Furthermore, the All incongruent condition should be the most attention demanding condition since there are two independent distractors, yet no regions show this in the contrast (Figure 3). Please comment on this. Is this true for the all incongruent vs. all congruent contrast?

5) Comparison with the behavioral data is based on the contrast between the AV congruent condition and the AV incongruent condition. Why did you not examine the contrast between AV congruent and the All incongruent condition? Was regression done on the contrast or was the contrast done on the regression results? It is not clear either why further analysis then relies on the AV congruent condition rather than the contrast. Were other conditions also analyzed?

6) The analysis is done purely in the spectral domain. It would also be of interest to know, e.g., the latency of the visual tracking effect and whether congruent visual information shortens the latency of auditory entrainment. Such analysis would help to link the current study with the previous literature on AV integration.

7) Although the relationship shown in Figure 5 between coherence in the left motor cortex and speech comprehension is very interesting, this does not show that the "motor cortex component contributes to speech comprehension" (Discussion), which would require demonstrating that disruption of activity in motor cortex affects speech comprehension.

8) Does the motor cortex activity include the lip regions? Please provide appropriate citations to justify this.

---

## [Author Response]

1) The reviewers felt that the paper would benefit from a stronger theoretical underpinning. At present, it is structured as an investigation into brain oscillations, which makes it hard to keep track of what are the main questions and findings. Some reorganization of the paper would help here.

We are grateful to the reviewers for the overall positive feedback to our manuscript. We have revised the logical foundation, organization and presented analyses within the revised manuscript according to the reviewer’s suggestions. Importantly, we now provide a stronger theoretical underpinning by opening the Introduction with an introduction to the multisensory phenomena related to lip reading. Then, we lead over to the potential relevance of brain oscillations in this context and outline the main four hypotheses that we test in this study. These four hypotheses provide a clear structure for the presentation of the analyses and the discussion.

2) The claim made at several points in the manuscript that lip movements entrain activity in auditory cortical areas does not appear to be justified. Although increased coherence was found there in the AV congruent vs. All incongruent condition, indicative of an interaction between the two modalities, lip movements in the absence of auditory speech (Figure 2—figure supplement 2) or with the contribution of speech signals removed (Figure 4) do not entrain auditory cortical activity. Thus, the effects on auditory cortex are only present during speech. The use of a partial correlation analysis to demonstrate that neural entrainment to lip movements is not an artifact caused by neural entrainment to the sound is very important and should be used throughout, not just toward the end of the Results.

We agree with the reviewers that we currently have no direct evidence that lip movements entrain auditory brain areas. Although coherence between lip movements and right auditory cortex increases significantly for AV congruent compared to All congruent condition, similar effects are absent in the partial coherence analysis. In response to the reviewers comment we have removed this claim from the revised manuscript. Instead we have added a short speculative comment about the coherence increase (AV cong > All cong) to the Discussion section. The relevant part reads:

“While visual areas show both coherence and partial coherence to lip movements, temporal areas only emerge from coherence analysis (Figure 2 and Figure 3—figure supplement 1). […] However, the informativeness of lip movements still modulates significantly the coherence between lip movements and right temporal brain areas (Figure 3—figure supplement 1) and indicates an indirect effect of visual attention.”

We have also addressed the reviewers’ comments that partial coherence should be used throughout. Specifically, we have moved Figure 3 (coherence results) to Supplemental Materials (Figure 3—figure supplement 1) and directly focus on the partial coherence results in the main analysis. Still, we believe that traditional coherence maps convey essential information and this information is still available as Supplemental Materials.

3) Figure 2 shows results at 1 Hz while Figure 3 shows the contrast at 4 Hz. It is stated that Figure 2 focused on 1 Hz since significant entrainment was seen "especially at 1 Hz". Does that mean stronger coherence or broader activation? It would be helpful to show the coherence spectrum of representative areas or to show the source distribution at several frequencies. Additionally, for the source-space plots, showing the differences between the actual coherence values could be more informative.

We now realise that the statement ‘especially at 1 Hz’ is misleading. This frequency was selected because the lip movement signal shows the strongest power at this frequency (Figure 2). The power spectrum for sound speech signals (for all the auditory stimuli we used) also shows the peak at 1 Hz as shown in Figure 5.

Author response image 1.**DOI:**
http://dx.doi.org/10.7554/eLife.14521.012

We have now rephrased the corresponding part in the revised manuscript:

“We first compared natural audiovisual speech condition (All congruent) and surrogate data for the frequency that showed strongest power in the lip signal (1 Hz). This revealed a significant entrainment effect in visual, auditory, and language areas bilaterally (*P* < 0.05, false discovery rate (FDR) corrected; Figure 2).”

In addition, as the reviewers suggested, we have added source distribution at several frequencies corresponding to Figure 2 as supplemental figures (Figure 2—figure supplement 3).

Figure 6 shows maps with actual coherence values for Figure 2.

Author response image 2.**DOI:**
http://dx.doi.org/10.7554/eLife.14521.013

In Figure 6 we also show maps of actual coherence value for each condition corresponding to Figure 3—figure supplement 1. However, we think the graphs in the original manuscript (*t*-statistics) are more informative than the plots in Figure 7.

Author response image 3.**DOI:**
http://dx.doi.org/10.7554/eLife.14521.014

4) Visual attention was not explicitly manipulated in this study. It is assumed that visual attention is enhanced when a competing sound is presented. Although this is a reasonable interpretation, it is not appropriate to report the auditory masking effect quite so definitively as a visual attention effect. Furthermore, the All incongruent condition should be the most attention demanding condition since there are two independent distractors, yet no regions show this in the contrast (Figure 3). Please comment on this. Is this true for the all incongruent vs. all congruent contrast?

We fully agree that we have not manipulated visual attention explicitly but only indirectly through the use of a competing sound. We have now clarified this point in the Discussion section of the revised manuscript. The relevant part reads:

“We studied the effect of attention in an ecologically valid scenario with congruent audiovisual speech when a distracting auditory stimulus was present (AV congruent condition) or absent (All congruent condition). […] This condition was compared to the All congruent condition where attention to the visual stimulus is not required for speech comprehension.”

We believe that the All incongruent condition is not the most attention demanding condition because participants were instructed to always attend to auditory stimuli (see Figure 1 where red letters represent attended stimulus). So there is no benefit for attending to the visual input in the All incongruent condition. This is different for the AV congruent condition where the congruent visual input facilitates comprehension of the attended auditory stream in the presence of another distracting auditory stream. Amongst all conditions the visual input is therefore likely most relevant and informative in the AV congruent condition. We have therefore used this condition for analysis of the ‘visual attention’ effect.

5) Comparison with the behavioral data is based on the contrast between the AV congruent condition and the AV incongruent condition. Why did you not examine the contrast between AV congruent and the All incongruent condition? Was regression done on the contrast or was the contrast done on the regression results? It is not clear either why further analysis then relies on the AV congruent condition rather than the contrast. Were other conditions also analyzed?

The analysis of the behavioural relevance of lip entrainment results was based on the rationale that this effect should be strongest when we compare our main condition (AV congruent) to the condition with the largest difference in behavioural performance, which is the AV incongruent condition (see Figure 1). We performed the regression analysis for each condition and then t-values at each brain voxel from the regression analysis were transformed to standard z-values to be compared between conditions. Then the z-values were subtracted between the two conditions. We have now described the procedure in more detail in the Methods section of the revised manuscript. The relevant section reads:

“To study the relationship between lip movement entrainment and behavioral performance, we performed regression analysis across subjects using comprehension accuracy from each individual as regressors on the partial coherence map. […] Then the *Z*-values were subtracted between the two conditions (*P* < 0.005).”

The contrast between AV congruent and AV incongruent condition already establishes a correlation between lip-entrainment and behavioural performance. Yet, to obtain further insights into this relation we performed additional analysis on the AV congruent condition, as it is the condition where the visual input is most relevant and informative for comprehension.

6) The analysis is done purely in the spectral domain. It would also be of interest to know, e.g., the latency of the visual tracking effect and whether congruent visual information shortens the latency of auditory entrainment. Such analysis would help to link the current study with the previous literature on AV integration.

Thank you for raising this important and interesting point and we have looked at the latencies of visual and auditory tracking in each condition and the effect of congruence.

We first checked the lag between lip (visual) and sound (auditory) speech signals. We performed cross-correlation analysis to estimate the delay between two signals. We computed this cross-correlation for both matching and non-matching signals used in the study. As shown in Figure 8, visual signals lead auditory signals by about 70 ms (here negative latency indicates lip speech (visual) signals lead sound speech (auditory) signals) for matching signals, however as expected there’s no effect for non-matching signals.

Author response image 4.**DOI:**
http://dx.doi.org/10.7554/eLife.14521.015

Next, we performed the cross-correlation analysis between speech signals and brain responses to measure visual and auditory tracking lag and how they change across conditions. We computed cross-correlation for both lip speech-visual cortex signals (visual tracking) and sound speech-auditory cortex signals (auditory tracking). Here the visual and auditory cortices were selected from the maximum voxels in the Figure 2 (left visual cortex (MNI coordinates = [-28 -88 8]), right STG (MNI coordinates = [44 -32 16])).

In Figure 9 the latency indicates the lag between signals in visual or auditory cortex and the lip or sound speech signals (brain signals following the speech signals). The mean lag across conditions is around 100 ms for all conditions as can be expected from studies of temporal response functions (e.g. Crosse et al., 2015). Congruent visual speech reduces the mean lag of auditory entrainment as can be seen in the comparison of ‘All cong A’ and ‘All incong A’. This is consistent with previous findings (van Wassenhove et al., 2005). But the difference is not significant and performing this analysis on narrow-band (filtered) data is not without problems. We therefore prefer to address this question in a separate study.

Author response image 5.**DOI:**
http://dx.doi.org/10.7554/eLife.14521.016

7) Although the relationship shown in Figure 5 between coherence in the left motor cortex and speech comprehension is very interesting, this does not show that the "motor cortex component contributes to speech comprehension" (Discussion), which would require demonstrating that disruption of activity in motor cortex affects speech comprehension.

We agree with the reviewers that this statement was too strong since we did not demonstrate a causal contribution of activity in the left motor cortex to comprehension. We have therefore rephrased this by stating “entrainment in motor regions correlates with speech comprehension” (Discussion).

8) Does the motor cortex activity include the lip regions? Please provide appropriate citations to justify this.

The area in the motor cortex showing significant lip-brain coherence (now Figure 3—figure supplement 1) is consistent with the lip/tongue representation in motor studies (see for example Giraud et al., 2007; Simonyan and Horwitz, 2011). In addition, a second area in left motor cortex emerged from the partial coherence analysis. This area was found in upper strip of primary motor cortex (maximum MNI coordinates in the partial coherence results for congruence effect (AV cong > All incong in Figure 3): [-52 -8 56]; for attention effect (AV cong > All cong in Figure 3: [-44 -16 56]; both clusters are extending over 10 mm radius). This is consistent with our previous finding from a different study that the same area exerts top-down control of left auditory cortex at the theta frequency band (Park et al., 2015). Although this second area appears to be located more superior compared to the classical lip/tongue motor representation, it still overlaps with motor areas that have been reported in fMRI studies of lip movements (Morillon et al., 2010; Table 1 in Fukunaga et al., 2009; lip panel in Figure 2 in Lotze et al., 2000) and articulatory movements of the lips (Figure 3 in Pulvermuller et al., 2006; this region was found to have causal influence on comprehending single spoken words in the TMS study (Schomers et al., 2015)). Further, this area is also consistent with areas observed in the study demonstrating essential role of motor cortex by disrupting it with TMS which leads to impaired speech perception (phonetic discrimination task) (Table 1 in Meister et al., 2007). Thus, we think that the motor cortex found by partial coherence analysis plays an important role in speech perception as we suggested in the Discussion (subsection “Motor areas are entrained by speaking lips”).